# Energy dissipation on magic angle twisted bilayer graphene

Alexina Ollier [1,2,5✉], Marcin Kisiel [1,5✉], Xiaobo Lu [3], Urs Gysin[1], Martino Poggio [1,2], Dmitri K. Efetov [4] & Ernst Meyer [1]

Traditional Joule dissipation omnipresent in today's electronic devices is well understood while the energy loss of the strongly interacting electron systems remains largely unexplored. Twisted bilayer graphene (tBLG) is a host to interaction-driven correlated insulating phases, when the relative rotation is close to the magic angle (1.08°). We report on low-temperature (5K) nanomechanical energy dissipation of tBLG measured by pendulum atomic force microscopy (p-AFM). The ultrasensitive cantilever tip acting as an oscillating gate over the quantum device shows dissipation peaks attributed to different fractional fillings of the flat energy bands. Local detection allows to determine the twist angle and spatially resolved dissipation images showed the existence of hundred-nanometer domains of different doping. Application of magnetic fields provoked strong oscillations of the dissipation signal at 3/4 band filling, identified in analogy to Aharonov-Bohm oscillations, a wavefunction interference present between domains of different doping and a signature of orbital ferromagnetism.

[1] Department of Physics, University of Basel, Klingelbergstrasse 82, CH-4056 Basel, Switzerland. [2] Swiss Nanoscience Institute, Klingelbergstrasse 82, CH-4056 Basel, Switzerland. [3] International Center for Quantum Materials, Collaborative Innovation Center of Quantum Matter, Peking University, Beijing 100871, China. [4] Department of Physics, Ludwig-Maximilians-University München, Geschwister-Scholl-Platz 1, 80539 München, Germany. [5] These authors contributed equally: Alexina Ollier, Marcin Kisiel. ✉email: alexina.ollier@unibas.ch; marcin.kisiel@unibas.ch

Twisted bilayer graphene (tBLG) at the magic angle twist ($\theta \approx 1.08°$), has gained a lot of interest due to the increase of Coulomb repulsion and the existence of flat energy bands close to Fermi level[1] leading to the emergence of exotic quantum phases such as superconductivity[2–5], ferromagnetism[6,7] and superlattice induced correlated insulating states (SIS)[3,5,8–10]. The recent advances in microfabrication enable to obtain high quality tBLG with energy spectra revealing features such as van Hove singularities[11,12], Hofstadter butterfly spectrum[6], WSe2 proximity induced spin-orbit interaction[13,14] or recently reported existence of magnetism in orbital Chern insulator[15,16]. Two layers of graphene, when stacked on top of each other, create a superlattice structure called Moiré pattern that can be filled with four electrons ($n_s = 4$). The interlayer hybridization effects open an energy gap at the $\Gamma$ point of the mini Brillouin zone (mBZ) and lead to drastic reduction of Fermi velocity and in consequence emergence of the flat superlattice minibands. The miniband filling is given by the band filling ($\nu$) or filling factor(FF) and is equal to $\nu = n/n_s = 1/4, 2/4, 3/4$, where $n$ stands for charge density per mBZ. During the last decade SIS in tBLG were extensively studied especially in electrical transport conductivity[3,9], thermal conductivity[17,18] and capacitance spectroscopy measurements[9,19]. Low-temperature transport measurements[2,3,9,20–22] showed a series of conductance drops, whereas half-filling of the mBZ leads to a drop of the quantum capacitance of tBLG device detected by a low-temperature capacitance bridge[9,19]. Many devices were studied[23] and experiments showed that surface imperfections, twist angle relaxations, and parasitic resistance of the electrically contacted samples is crucial to obtain a whole series of superconducting, correlated, and magnetic states in tBLG. Therefore, transport and capacitance measurements require clean and homogeneous samples due to the detection method that averages over the whole device. The averaging effect is less in local probe measurements, due to the detection at specific surface spots. The SIS of tBLG were reported by scanning tunneling microscopy (STM)[24–27], SQUID-on-tip[15,16,22] and tuning fork atomic force microscopy (AFM)[28]. Except SQUID experiments, all other scanning probe measurements structures had essentially open surface i.e., the tip operated in close proximity to the non-encased tBLG device.

While dozens of groups worldwide are focused on various electronic and structural properties of tBLG, our aim is to study the nanomechanical dissipation with very sensitive AFM tips. Breakdown of topological protection[29], loss of quantum information, and disorder-assisted hot electrons scattering in graphene[30] are just few examples of systems, where the presence of energy dissipation has a great impact on the studied object. It is therefore critical to know, how and where the energy leaks.

In this contribution we use non-contact pendulum geometry AFM to detect the series of SIS in tBLG by purely mechanical means. Owing to high force sensitivity (see "Methods" section), pendulum geometry Atomic Force Microscope (p-AFM), oscillating like a tiny pendulum over the surface, is perfectly suited to measure tiny amount of energy loss[31,32]. The measurements reveal the rise of mechanical dissipation at half-filling, as well as for $\nu = \pm 1/4, \pm 3/4$, and 4/4. In the present non-contact realization, the experiments are conducted at large distance ($d = 150$ nm) between the tip and the sample with tip oscillating at extremely low frequency of 13 kHz. Moreover, SIS are detected above an encapsulated device, therefore the tip literally does couple to the existing subsurface phenomena. The rise of dissipation signal is related to the creation of displacement currents under the oscillating tip as well as the change of quantum capacitance of the sample when the charges are injected into the flat energy bands. Both phenomena affect the dynamics of the oscillating tip and lead to the rise of mechanical damping of the cantilever. It is known

that the superlattice density $n_s$ depends on the twist angle $\Theta$ as following: $n_s = \frac{8\Theta^2}{\sqrt{3}a^2}$, where $a$ is the lattice constant of graphene[3]. Since the electronic properties of tBLG are extremely sensitive to the homogeneity of the twist angle the cautious control of the stacking process and subsequent cleaning are crucial[23]. Owing to the local character of the measurement, the sharp tip of p-AFM positioned at different sample spots is able to confirm high quality of tBLG devices and to determine a narrow twist angle distribution equal to $\Delta\Theta = 1.06° \pm 4\%$ over micrometer distances. The dissipation spectra and the constant height dissipation images were acquired, showing the existence of few hundred-nanometer domains of different local doping, which is confirmed by a spatial variation of the charge neutrality point (CNP).

Application of magnetic fields leads to strong oscillation of the energy dissipation signal which is enhanced for fractional 3/4 band filling. The magnetic field (B) induced oscillations appear at different B-fields and show few B-field periodicities. We discriminate two types of magneto-oscillations. The oscillations observed at larger magnetic fields with small periodicity are identified as originating from quantum interference effects occurring at boundaries between domains of different doping. The phenomena is commonly known as Aharonov-Bohm effect. The observed oscillations with larger periodicity, localized near zero B-field are consistent with recent SQUID-on-tip measurements, which supports the presence of orbital magnetism reported recently in tBLG devices at 3/4 band filling[15,16].

## Results

**Nanomechanical dissipation from correlated insulating states.** The schematics of the p-AFM oscillating on top of the tBLG device is shown in Fig. 1a and the details are given in the "Methods" section and Supplementary Note 1. The structures were etched into Hall geometry for initial transport experiments[23]. The sample consists of a p-doped silicon substrate with a 300 nm thick silicon dioxide (SiO2) and 10 nm thick hexagonal boron nitride (hBN). The tBLG is capped from top with a 10nm thick hBN that prevents tBLG from contamination. Since the dielectric constant of hBN is equal to $\epsilon = 4$, 10 nm of hBN is equivalent to 40 nm of vacuum gap between tip and sample. The cantilever is oscillating with a fixed amplitude of $A = 1$ nm, at a fixed tip-sample distance $d = 150$ nm, whereas a backgate voltage is applied to the silicon substrate. The cantilever is grounded whereas the potential of tBLG is controlled with two gold electrodes (see the "Methods" section). In most of the experiments, we kept tBLG grounded as shown in Fig. 1a. Figure 1b shows the non-contact p-AFM image of the tBLG device.

To investigate the energy loss mechanisms, we performed dissipation and force spectroscopy measurements. While the backgate voltage ($V_{BG}$) was swept from $V_{BG} = -50$ V to 60 V the dissipation $\Gamma$ and the frequency shift $\Delta f$ spectra were simultaneously recorded. The typical $\Gamma$ spectrum versus charge concentration $n$ is presented in Fig. 1c. The details about the conversion of $V_{BG}$ to $n$ are given in Supplementary Note 2. The dissipation curve shows a series of peaks that are symmetric with respect to the charge neutrality point. Owing to disorder introduced by charge impurities, the data (see Supplementary Note 2, Fig. S3 and Supplementary Note 3, Fig. S4a) show the contact potential difference (CPD) from charge neutrality point equal to $V_{CPD} = 8$ V. The corresponding disorder density is equal to $n_d = 5 \cdot 10^9$ cm$^{-2}$, which is in good agreement with values already reported[9]. The positions of the dissipation peaks exactly corresponds to the mBZ filling factor $\nu$ and the $\Gamma$ rise is observed for band filling equal to $\nu = 1/4, 2/4, 3/4$ and 4/4. Based on the measured spectra we determine the twist angle $\Theta = 1.08°$ and the corresponding superlattice density $n_s = 2.6 \cdot 10^{12}$ cm$^{-2}$, both

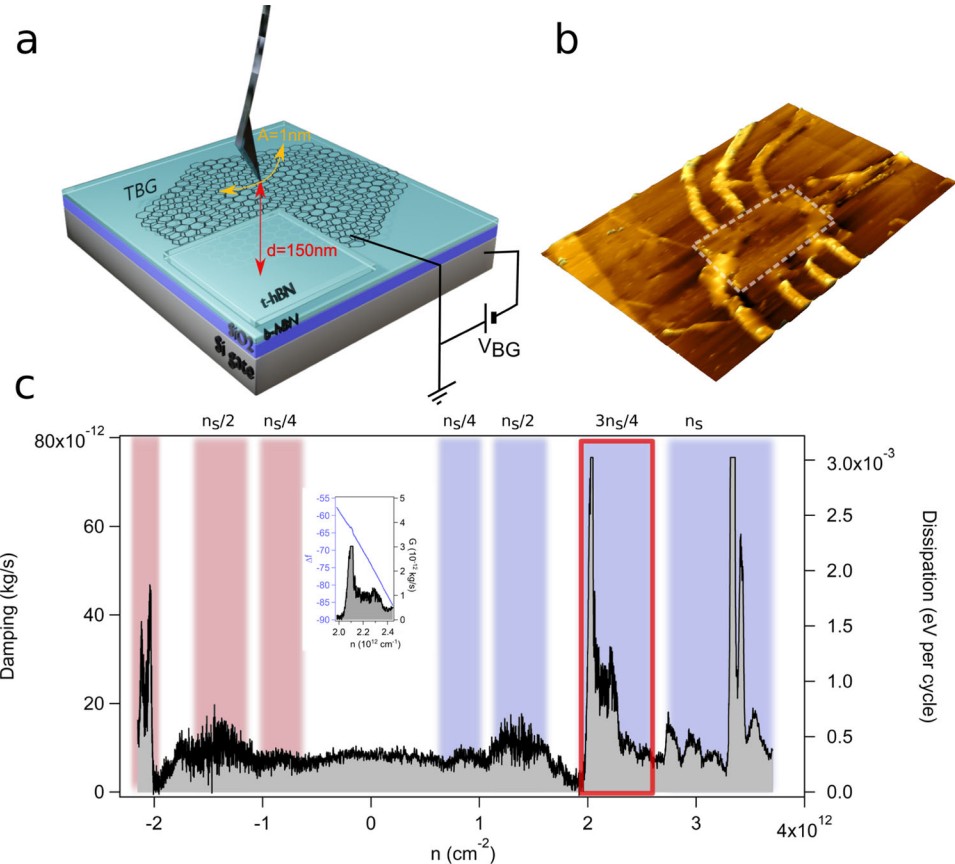

**Fig. 1 Pendulum atomic force microscope (p-AFM) tip oscillating on top of a twisted bilayer graphene (tBLG) device is measuring energy loss. a** schematics of the measured device. The sample is composed of a p-doped silicon backgate (dark gray), followed by a 300 nm thick silicon dioxide layer (violet) and 10 nm hexagonal boron nitride (hBN, light blue). From the top tBLG is encapsulated with 10nm hBN film. **b** p-AFM image of the tBLG device, the white dotted rectangle is the device with a size of 4.1 μm × 5.8 μm. In **c** the typical dissipation (Γ) spectrum versus doping concentration (n) is shown. Different peaks in Γ spectrum are identified as $\nu = \pm 1/4$, $\nu = \pm 1/2$, $\nu = \pm 3/4$, and $\nu = 4/4$ and colored in red and blue for holes and electrons, respectively. The inset shows the corresponding frequency shift $\Delta f(3n_s/4)$ spectrum. The measurement was performed at temperature $T = 5$ K.

values corresponding to magic angle twist[3,9]. We noticed that the dissipation peaks of high intensity observed for $\nu = 3/4$ and $4/4$ filling factors are accompanied with a tiny change of the $\Delta f$ signal, as visible in the inset in Fig. 1c.

**Twist angle distribution**. Due to local character of the method, p-AFM is perfectly suited to determine the twist angle at different sample locations. Thus, we acquired the energy dissipation Γ spectrum at different positions of the device and we match the position of the observed dissipation peaks to the superlattice density $n_s$. 55 subsequent dissipation spectra were acquired along the 1 μm long line (red line in Fig. 2a and Supplementary Note 4) and for each spectrum both the twist angle Θ and CPD shift from charge neutrality point were determined. Both data are shown in Fig. 2b. A weak cross correlation (10%) as a function of displacement was noticed between Θ and CPD values, meaning both observables are not inherently linked. Figure 2c is a histogram of the Θ angle and the red dotted line is a Gaussian fit that reveals a mean twist angle equal to $\Theta = 1.06°$ with an inaccuracy of about 4%, which demonstrates a decent twist angle homogeneity. Although we noticed good twist angle homogeneity, a presence of charge disorder leads to considerable variation of CPD values. Moreover, CPD data sets (Fig. 2b) has both negative and positive values, suggesting the presence of p- and n-doped regions on the surface. Thus, the data suggest the presence of domains of different local doping, as expected for van der Waals heterostructure supported on $SiO_2$ substrates[33].

In order to further investigate those domains a constant height $\Gamma(V_{BG})$ images were acquired as shown in Fig. 2d, e. The grounded tip was positioned at a distance $d = 150$ nm above the surface and $V_{BG} = 24$ V and 31.4 V was applied to the backgate. Those values correspond to dissipation peaks visible in Fig. 1c and are labeled with filling factor $\nu = 1/4$ and $2/4$, respectively. Thus, the dissipation contrast is solely from the position change of the Γ peak due to local variation of Θ and CPD value. Due to high twist angle homogeneity, most of the dissipation contrast originates from CPD variations (see Fig. 2b). Both maps reveal the presence of domains having sizes of few hundred nanometers (marked with white dotted lines). The corresponding maps of $\Delta f$ are shown in Supplementary Note 5. Circular features observed in Γ images are presumably due to single electron charging - a phenomenon already reported in AFM measurements[34–36] and known as Coulomb blockade (CB). In local probe measurements single electron charging is visible as spatially extended Coulomb rings separating the different charge states of the quantum-dot-like entity. It is not clear whether the source of CB could be a point defect in graphene or hBN. It could also be an imperfection introduced into tBLG during stacking process. It was demonstrated that out of plane deformed graphene might behave as an effective quantum dot[37]. No significant change of contrast of hundred-nanometer domains was observed between images taken at different $V_{BG}$, whereas CB rings show strong voltage dependence, which suggests different origins of both features.

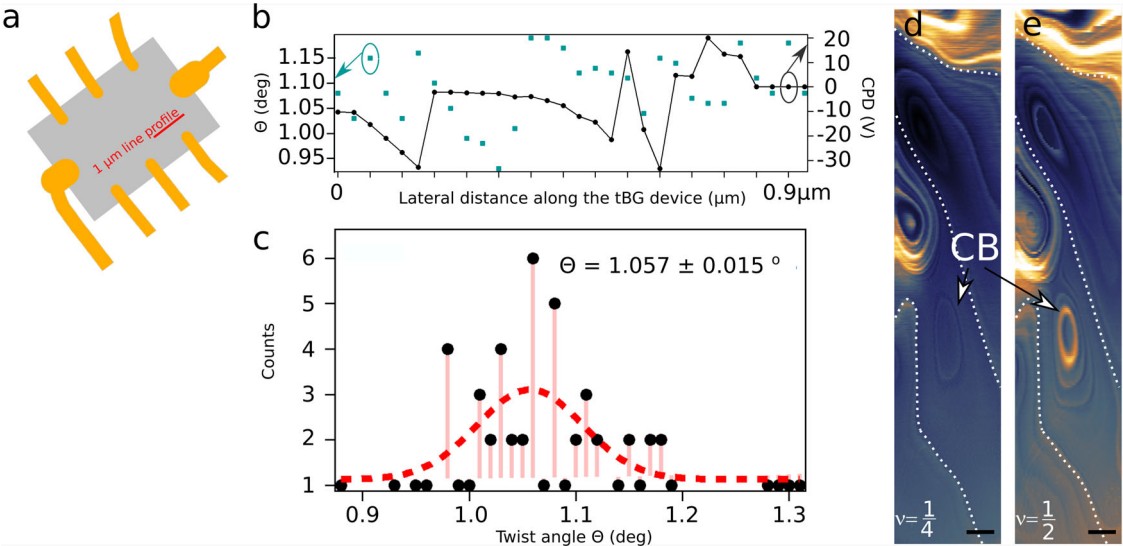

**Fig. 2 Spatial distribution of twist angle Θ and Contact Potential Difference (CPD) from the charge neutrality point (CNP). a** schematics of the measured device. To determine the twist angle Θ, fifty five dissipation spectra were acquired along the 1 μm long red line. **b** twist angle Θ (green) and CPD (black) measured along the line profile shown in (**a**). A variation of CPD value is noticed and is suggesting a certain degree of charge disorder. In **c** the histogram of the determined Θ values is shown, where the red dashed line is a Gaussian fit to the measurement data. The twist angle is equal to Θ = 1.057° ± 0.015°, indicating minimal effects of angle variation. **d, e** constant height ($d = 150$ nm) dissipation maps taken for band fillings $\nu = 1/4$ and $\nu = 1/2$, respectively. The corresponding backgate voltages $V_{BG}$ were equal to 24V and 31.4 V for $\nu = 1/4$ and $\nu = 1/2$, respectively. Domains having sizes of a few hundred nanometers are clearly visible and are marked with white dashed lines. The domains emerge due to spatial variation of both Θ and CPD. A round shaped dissipation contrast is presumably due to Coulomb blockade (CB) effect. The scale bar is 50 nm.

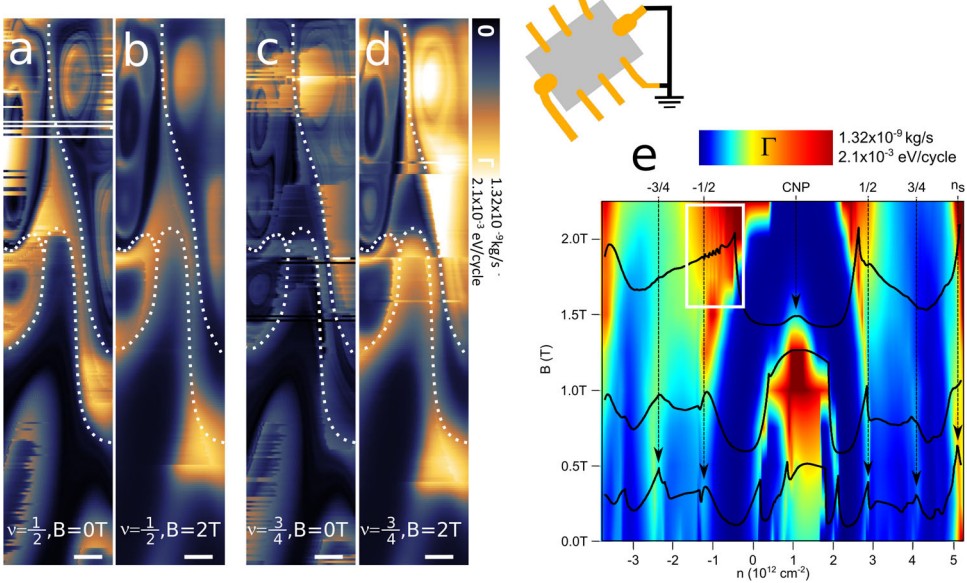

**Fig. 3 Constant height ($d = 150$ nm) dissipation images and spectra versus doping ($n$) and magnetic field (B) while the twisted bilayer graphene (tBLG) device was kept grounded. a–d** dissipation (Γ) contrast for band filling $\nu = 1/2$ (**a, b**) and $\nu = 3/4$ (**c, d**) without external B-field applied (**a, c**) and with external magnetic field $B = 2$ T (**b, d**) applied in a direction perpendicular to the sample surface. Hundred-nanometer-sized domains are visible and marked with white, dashed lines. Slight rise of Γ intensity was noticed under application of B-field. The scale bar is 50 nm. In **e** the Γ intensity map and spectra are shown versus $n$ and B-field. Application of B-fields equal to about $B = 2$ T leads to emergence of magneto-oscillations (marked with white rectangle).

**Dissipation oscillations under external magnetic field**. Next, we applied an external magnetic field in direction perpendicular to the sample surface. Figure 3a–d are constant height Γ images for $\nu = 1/2$ and $\nu = 3/4$, respectively. The images were taken at different sample spots as compared to Fig. 2d, e. While images in panels (a) and (c) show the data for $B = 0$ T, the data in panels (b) and (d) are taken with applied external B-field equal to $B = 2$ T.

All data again show the presence of hundred-nanometer-sized domains that are highlighted with white, dashed lines. Furthermore, we noticed that the domain contrast, especially for $\nu = 3/4$ filling, is enhanced under application of B-fields (see Fig. 3c, d). The Γ intensity map with few superimposed dissipation spectra versus charge density $n$ and B-field is plotted in Fig. 3e and also shows an increase of Γ versus B-field. The observed enhancement

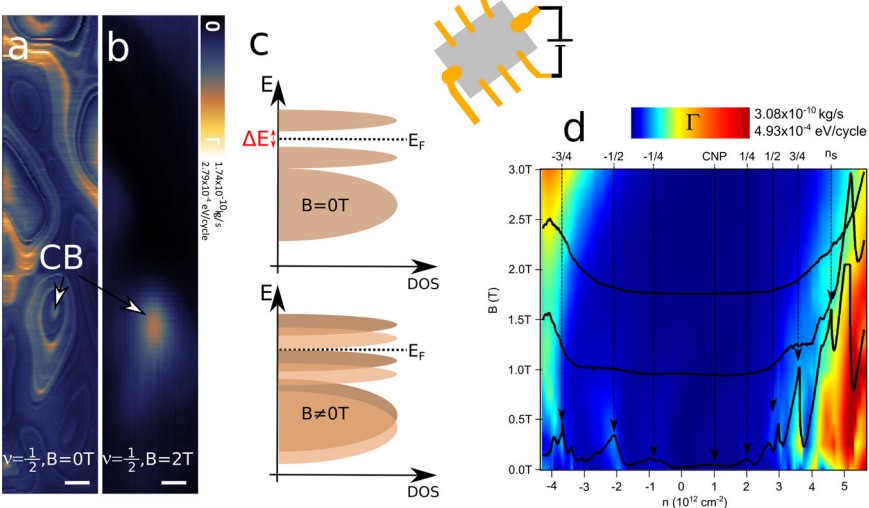

**Fig. 4 Constant height ($d = 150$ nm) dissipation images and spectra versus doping ($n$) and magnetic field (B) while small in-plane voltage equal fractions of millivolt was applied to the twisted bilayer graphene (tBLG) device. a** dissipation ($\Gamma$) contrast for band filling $\nu = 1/2$ without external B-field applied. Hundred-nanometer-sized domains as well as Coulomb blockade (CB) features are visible. Image **b** is taken under the same conditions as (**a**), yet with applied the external magnetic field equal to $B = 2$ T. Although the $\Gamma$ contrast from hundred-nanometer domains disappeared, some contrast from CB single electron charging is still present. The scale bar is 50 nm. **c** possible mechanism of energy gap closing due to B-field induced Zeeman shift (see the "Dissipation oscillations under external magnetic field" section). On **d** the $\Gamma$ intensity map and spectra are shown versus $n$ and B-field. Application of B-fields larger than $B = 2$ T leads to disappearance of the dissipation peaks.

of dissipation contrast at CNP for $B = 1$ T is presumably due to increased electron-hole scattering when an external B-field drives charges in the opposite direction. At charge densities $1/2 < \nu < 3/4$ and for non-zero B-field we observed the oscillations in $\Gamma$ signal (marked with white rectangle in Fig. 3e), which we further analyzed in details, yet before we discuss it, we would make an important comment about dissipation behavior versus B-field. Until now, we reported on the measurements with the grounded cantilever positioned over the grounded tBLG device as it is shown in Fig. 1a. In the presence of small electric field in order of mV/μm applied in plane of the tBLG sample (see inset in Fig. 4), $\Gamma$ contrast evolution versus B-field is different. Figure 4a, b are the dissipation images for $\nu = 1/2$ taken under no B-field applied and under B-field equal to $B = 2$ T, respectively. In contrast to the case of grounded tBLG, the application of magnetic fields leads to strong $\Gamma$ contrast reduction. The hundred-nanometer-sized domains vanish, whereas we observe some remnant contrast of CB rings, which again confirms that both features are of different origin. The dissipation spectra versus $V_{BG}$ under external B-field are shown in Fig. 4d. Again, we point out that $\Gamma$ contrast reduction versus B-field was observed only in the presence of small in-plane electric field applied to tBLG.

**Magneto-oscillations at $\nu = 3/4$ band filling**. To corroborate on magneto-oscillations (Fig. 3e), the energy dissipation response of the grounded tBLG was further studied under varying B-field. The tip-sample distance ($d = 150$ nm) and $V_{BG}$ were set constant and the B-field was swept from $-2.5$ T $< B < 2.5$ T. The sweep rate was equal to 0.07 T per minute. $\Delta f(B)$ and $\Gamma(B)$ spectra were acquired for subsequent band fillings equal to $\nu = 1/4$, 1/2, 3/4 and for the filling slightly above, namely $3/4 < \nu < 4/4$. The corresponding charge concentrations were equal to $0.9 \cdot 10^{12}$ cm$^{-2}$, $1.4 \cdot 10^{12}$ cm$^{-2}$, $2.1 \cdot 10^{12}$ cm$^{-2}$ and $2.3 \cdot 10^{12}$ cm$^{-2}$, respectively (see Fig. 1c). The recorded dissipation $\Gamma$ versus B spectra are shown in Fig. 5a–c. The dissipation signal shows oscillations that are strongly enhanced for $\nu = 3/4$ and $3/4 < \nu < 4/4$. Analogous magneto-oscillations are also observed in $\Delta f(B)$ spectra and those data are shown in Supplementary Note 6, Fig. S7. Whereas half-

filling $\nu = 1/2$ spectrum shows only residual oscillations, the data for $\nu = 3/4$ are characterized by two different types of oscillations, namely those localized at large B-field ($B > |2$ T$|$) and those present at low B-field ($B < |0.1$ T$|$). Further increase of band filling to $3/4 < \nu < 4/4$ results in continuous magneto-oscillations all along the $\Gamma(B)$ spectrum, as shown in Fig. 5c. The insets show at least two different periodicities at $B = 0$ T and $B = 1$ T. Next, we analyzed the Fast Fourier Transformed (FFT) dissipation spectra shown in Fig. 5d–f and Fig. 5g–i for band fillings equal to $\nu = 3/4$ and $3/4 < \nu < 4/4$, respectively. Both filling factors (FF) revealed a distribution of periodicities as seen in Fig. 5d, g with $1/B$ aperiodic character which excludes the presence of Shubnikov-de-Haas oscillations due to emergence of apparent Fermi surface as previously reported in bilayer graphene[3,38]. The FFT analysis was performed separately for low B-regions (Fig. 5e, h for $\nu = 3/4$ and $3/4 < \nu < 4/4$, respectively) and high B-regions (Fig. 5f, i for $\nu = 3/4$ and $3/4 < \nu < 4/4$, respectively). A distinction between oscillations periodicity for high (marked with green arrows) and low (marked with black arrow) B-regions is visible. At low B-regions the oscillations have periodicity $\Delta B_{low} \approx 16$ mT (Fig. 5e), whereas FFT spectrum taken at high B-regions and shown in Fig. 5f shows two peaks localized at $\Delta B_{high} \approx 5$ mT and 8 mT. A separation of high and low B-field oscillations visible in Fig. 5b as well as their different periodicities points to the different origins. At $3/4 < \nu < 4/4$ FF (Fig. 5h, i) similar peaks that are marked with green and black arrows are again visible. Moreover, the additional peaks appear (marked with triangles) and since their frequencies roughly match 16–5 mT and 16–8 mT, those are presumably due to beat interference between oscillations present at high and low B-regions. A last remark before closing the "Results" section. Although, $\Gamma(B)$ spectrum for $\nu = 1/2$ filling (Fig. 5a) is in most parts smooth, a tiny oscillations are visible at $B = 0$ T and at large B-fields, which indicate that although the tip mostly senses $\nu = 1/2$ domains underneath, a minute amount of domains with $\nu = 3/4$ filling is present in the tip proximity. Therefore, despite the decent twist angle homogeneity measured across micrometer distance, somewhere on the 'horizon' of the tip shadow potential must be domains of different angle twist or different doping.

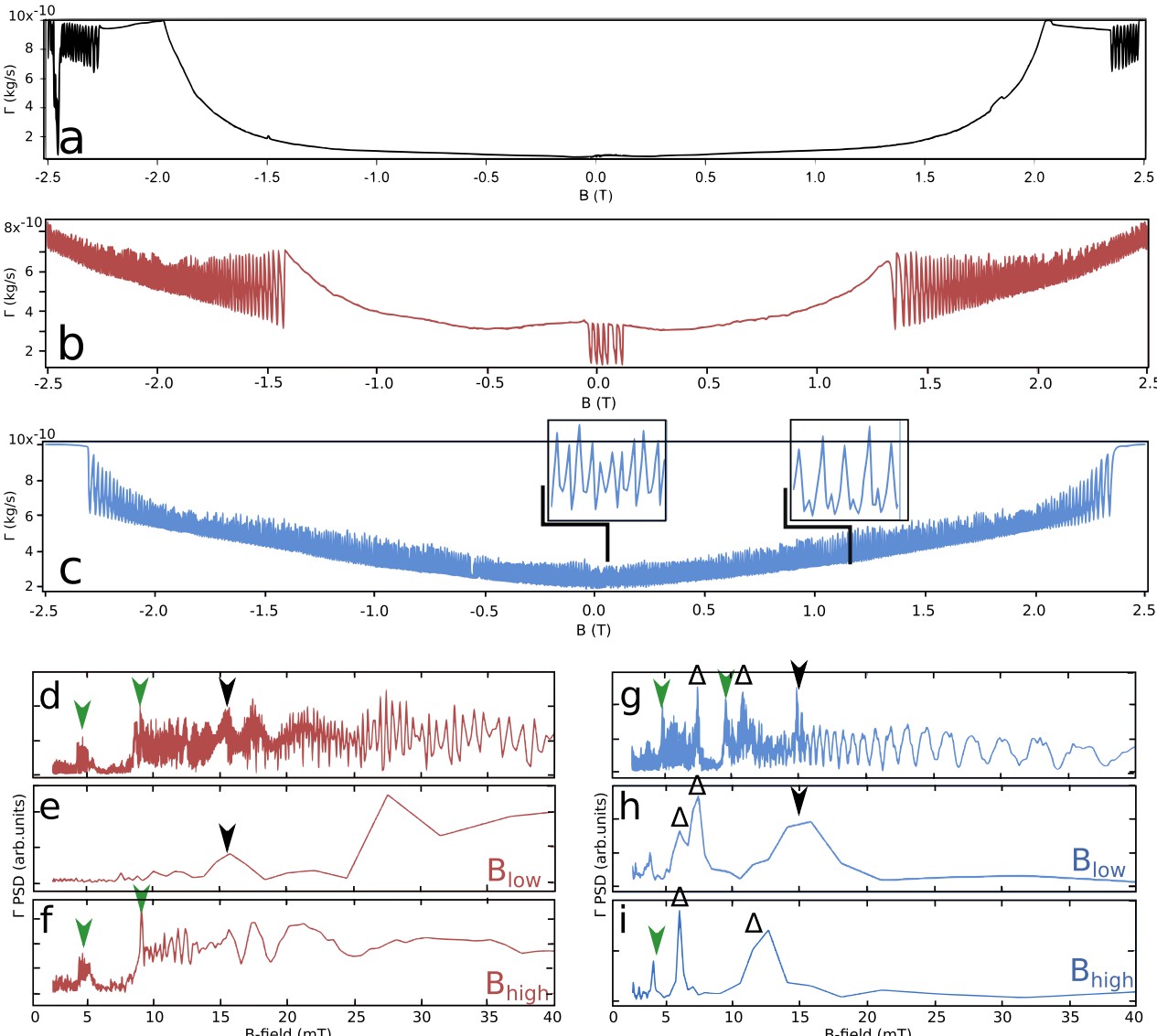

**Fig. 5 Constant height (d = 150 nm) dissipation (Γ) spectra versus magnetic field (B) for different band filling (ν). a–c** Γ versus B-field spectra for ν = 1/2, ν = 3/4 and 3/4 < ν < 4/4 band filling, respectively. **d–f** fast Fourier transform (FFT) at ν = 3/4 filling for full spectrum, low B-region and high B-region, respectively. **g–i** FFT at 3/4 < ν < 4/4 filling for full spectrum, low B-region and high B-region, respectively. Markers denote the characteristic frequencies of magneto-oscillations (see the "Magneto-oscillations at ν = 3/4 band filling" section).

Similar applies to the magneto-oscillation visible at 3/4 < ν < 4/4 FF, when the tip mostly senses p- or n-doped semiconducting domains resulting from close proximity of the full filling. Hence, the effective sample area probed by the oscillating pendulum tip at d = 150 nm distance (the dissipation versus tip-sample distance dependence is presented in Supplementary Note 7, Fig. S8.) must be in the order of few hundred nanometers. The reported domains size is in good agreement with SQUID-on-tip measurements[15] operating at similar tip-sample distance of 140 nm.

## Discussion
The p-AFM tip oscillating at hundred-nanometer distance from the tBLG device is coupled capacitively to the underlying tBLG and is sensitive to the series of SIS in both electron and hole doped regions. SIS are detected via mechanical energy loss of the cantilever sensor. Owing to large tip-sample distance, the dominant dissipation mechanism is Joule dissipation (see Supplementary Note 8) which is linked to the creation of local,

displacement currents by an oscillating tip and reads as follows: $P = \frac{1}{2}RA^2\omega^2(\Delta\phi)^2\left(\frac{\partial C}{\partial z}\right)^2$, where $A$, $\omega$ are the cantilever oscillation amplitude and frequency, respectively. $\Delta\phi$ is defined by tip and sample potential and is equal to the difference of their work functions. Joule dissipation consists of two dissipation channels: resistive ($R$) and capacitive $\frac{\partial C}{\partial z}$. Therefore, the observed series of SIS are due to both, the rise of sample resistivity in the current path and an abrupt change of system capacitance, which involves quantum capacitance ($C_Q$) of tBLG. The resistive part produces wide and less pronounced dissipation peaks as observed for ν = 1/4 and 2/4 band filling, whereas the capacitance change leads to creation of much sharper spikes (ν = 3/4 and 4/4). It does so due to charge injection into flat energy bands, or else, due to the presence of van Hove singularities in the electronic density of states. The crossing of Fermi energy with van Hove singularity results in a change of thermodynamic compressibility $\partial n/\partial\mu$ of tBLG, which reflects the ability of tBLG to absorb an amount of charge $n$ when changing the chemical potential $\mu$. This process leads to a rapid change of quantum capacitance $C_Q = A_L e^2 \cdot \partial n/$

$\partial\mu$, where $e$ is the elementary charge, $A_L$ is the lateral device area probed by the tip. The frequency shift of the cantilever signal can be approximated by $\Delta f = \frac{\omega_0}{8\pi k}(\Delta\phi)^2 \frac{\partial^2 C}{\partial z^2}$ (see Supplementary Note 8), where $\omega_0$ and $k$ are eigenfrequency and stiffness of the cantilever. Since $\Delta f$ solely depends on system capacitance (while other parameters are fixed) the charge injection into the flat energy bands results in $\Delta f$ change, as observed for $\nu = 3/4$ and $4/4$ band filling (see inset Fig. 1c and Supplementary Note 3). Particularly at full filling the change of $\Delta f$ is in agreement with already reported huge rise of thermodynamic compressibility[39].

While p-AFM confirmed decent twist angle homogeneity a substantial charge disorder introduced by the SiO$_2$ substrate was detected and led to creation of hundred-nanometer-sized domains of different doping. Those domains respond differently to the applied B-field, depending on their electrostatic gating. While in grounded graphene the domains survive the application of B-fields as high as $B = 2.5$ T, the $\Gamma$ domain contrast is diminished by B-field in non-grounded tBLG areas. The presence of SIS under non-equilibrium conditions should be further investigated, therefore we restricted the discussion to a short remark. The charge redistribution under application of in-plane electric fields builds an electric field between the graphene layers which in turn modifies the interlayer interaction. Based on the results we conclude that in-plane polarization weakens the interlayer coupling leading to slight reduction of energy gap. Further application of relatively weak magnetic fields of the order of $=2$ T introduces a Zeeman shift which fully closes the energy gap and recovers a normal metal state. Cao et al.[9] reported that application of B-fields of few Tesla leads to a Zeeman shift of the order of few hundred of $\mu eV$ that suppresses the half-filling states. Our results support this scenario. The external magnetic field provides Zeeman energy $2\mu_B B \approx 0.2$ meV needed to at least partially close the correlated insulator energy gap, hence the vanishing of dissipation peaks characteristic for different SIS and overall $\Gamma$ contrast reduction (see Fig. 4c). In general, states that occur near half-filling are less robust as they have much smaller energy scale when compared to SIS localized at larger band fillings. They are characterized by the band gap equal to about 0.3 meV which corresponds to critical temperature equal to $Tc = 3.5$ K[9]. Larger temperature of our measurement ($T = 5$ K) leads to the reduction of dissipation intensity for $\nu = 1/2$ visible in dissipation spectra in Fig. 1c, as well as the reduction of magneto-oscillations in Fig. 5a.

The experiment on grounded tBLG and under moderate B-fields distinguished two different types of magneto-oscillations characterized by different periodicities, with distribution falling in between $\Delta B = 5$ mT $- 15$ mT. The rise of dissipation signal (also domain contrast) observed at $\nu = 3/4$ FF and at high B-fields suggests that the origin of the oscillations is related to the presence of hundred-nanometer-sized domains of different doping. The presence of positive and negative CPD domains leads to formation of local p-n junctions. Further application of B-fields creates a directional edge currents flowing at the junction boundaries and the quantum mechanical interference at domain boundaries leads to magneto-oscillations. Thus, the oscillations in high B-regions originate from the Aharonov-Bohm effect in analogy to oscillations reported in high-mobility GaAs two-dimensional electron gas[40,41] and recently reported in graphene based quantum Hall systems[42]. The period of Ahoronov-Bohm oscillations should satisfy well the standard formula: $\Delta B \cdot S = \frac{h}{e}$, where $\frac{h}{e}$ is the magnetic flux quanta and $S$ is the surface area enclosed by circulating channels inside the graphene domain. The measured periodicity $\Delta B = 10$ mT (see Fig. 5), results in domain sizes roughly equal to $S = 600$ nm, which is in agreement with the p-AFM domain images (Fig. 2d, e). The decreasing oscillation amplitude with increasing magnetic field (Fig. 5b) indicates the vanishing coupling between edge states as they move further apart from each other at higher magnetic fields.

We will make one remark before closing. Larger periodicity $\Delta B = 15$ mT oscillations present at very narrow B-field window near zero ($-100$ mT $< B < 100$ mT) are consistent with recent SQUID-on-tip measurements[15,16] which reported the emergence of orbital magnetism in tBLG devices at 3/4 band filling. Width of the oscillations reported by p-AFM is in agreement with the width of the hysteresis loop of Hall resistance, which has been found to be slightly below $\pm 100$ mT (see Fig. 1a in[15]), as well as with the evolution of the coercive field versus $n$ (see Fig. 2c therein). According to Tschirhart et al.[15], when two materials of opposite Chern number are put in contact the emergence of chiral edge states is expected at the interface[43]. The observation of magneto-oscillations present at B-field as low as $B < 100$ mT and localized at $\nu = 3/4$ band filling supports the scenario of chiral edge currents present at domain boundaries, as observed by SQUID measurements. Disappearance of magneto-oscillations at $3/4 < \nu < 4/4$ FF indicates that network of topologically protected chiral channels is highly sensitive to doping concentration.

## Conclusion

Low-temperature p-AFM mechanical dissipation detected the series of SIS in tBLG as a function of carrier density and magnetic field. It does it without touching the crystal and couples to the subsurface effect in the encapsulated quantum device. The mechanism of dissipation rise at different band filling was discussed and is consistent with the creation of local currents below the oscillating tip as well as a rapid change of quantum capacitance of tBLG device. Due to the local detection the p-AFM could quantify the twist angle and doping at various sample spots and concluded charge disorder to be dominant. Spatially resolved dissipation images showed the existence of hundred-nanometer domains. Application of magnetic fields leads to magneto-oscillations of the dissipation signal which is enhanced at $\nu = 3/4$ band filling. We identify those oscillations as originating from Aharonov-Bohm quantum interference effect at domain's boundaries of different doping. Another type of oscillations near $-100$ mT $< B < 100$ mT magnetic field were found and presence of those is in agreement with scenario of Chern domain walls. Finally, we have demonstrated that mechanical oscillators can address quantum effects in energy dissipation.

## Methods

**Sample preparation**. The tBLG device was fabricated by step-by-step stacking process described in details elsewhere[23]. The hBN/tBLG/hBN/graphite stacks were exfoliated and assembled using a van der Waals assembly technique. First 10 nm thick hBN flakes were exfoliated on SiO$_2$/Si p-doped substrate of resistivity $\rho < 0.005$ $\Omega$cm. Next the separated graphene pieces were rotated manually and mechanical cleaning process was applied to release the local strain. Finally tBLG device was capped with 10 nm hBN. The charge concentration was controlled with a DC voltage applied to the p-doped Si backgate, whereas tBLG was coupled capacitively via $d_{SiO} = 300$ nm of SiO$_2$ oxide and $d_{hBN} = 10$ nm of hBN. The geometric capacitance of the backgate was equal to $C_{ox} = \frac{C_{SiO} \cdot C_{hBN}}{C_{SiO} + C_{hBN}} = 1.11 \cdot 10^{-4} Fm^{-2}$, where $C_{SiO/hBN} = \frac{\varepsilon_{SiO/hBN}\varepsilon_0}{d_{SiO/hBN}}$. A tBLG device was contacted with 8 golden wires and two of them were grounded during the measurement.

**Pendulum AFM energy dissipation measurements**. Series of insulating states of tBLG were detected under ultra high vacuum (UHV) conditions with highly n-doped silicon ATEC-Cont

cantilever from Nanosensors. The cantilever with resistivity $\rho = 0.01 - 0.02 \, \Omega cm$, spring constant $k = 0.18 \, N/m$ and frequency $f_0 = 13 \, kHz$ was coupled capacitively to the quantum device, and the tip oscillation amplitude $A = 1 \, nm$ was parallel to the sample surface (pendulum geometry). The sensor was annealed before experiment at 700° under UHV conditions for 12h. The process leads to removal of water and weakly bounded molecules from the cantilever surface and the tip. Moreover, the long term annealing minimizes the amount of the static charges localized at the tip. After the annealing the quality factor ($Q$) of the sensor is improved and equal to $Q = 550 \times 10^3$. The corresponding force sensitivity of the sensor is equal to $F_{min} = 2.34 \times 10^{-17} \, N/\sqrt{Hz}$. During the experiment the tip was electrically grounded. The oscillation frequency and the amplitude of the lever were controlled by means of phase locked loop (PLL) electronic circuit[44]. This detection method is sensing the change of the frequency shift ($\Delta f$) caused by the tip-sample interaction as a function of the applied bias voltage as well as the excitation voltage needed to keep the oscillation amplitude $A_{exc}$ constant. The non-contact dissipation is calculated according to:

$$\Gamma(d) = \Gamma_0 \left( \frac{A_{exc}(d)}{A_{exc,0}} - \frac{f(d)}{f_0} \right) \quad (1)$$

where $A_{exc}(d)$ and $f(d)$ are the distance-dependent excitation amplitude and resonance frequency of the cantilever, the index 0 refers to the free cantilever.

## Data availability
All the data are available upon request.

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

## Acknowledgements
Financial support from the Swiss National Science Foundation (SNSF grant 200020-188445), the SNSF through NCCR SPIN (SNSF grant 51NF40-180604), and the Swiss Nanoscience Institute (SNI) is gratefully acknowledged. We also thank the European Research Council (ERC) under the European Union's Horizon 2020 research and innovation program (ULTRADISS Grant Agreement No. 834402). We thank Prof. Dr. Erio Tosatti and Prof. Dr. Christian Schönenberger for fruitful discussions.

## Author contributions

E.M., M.P., and M.K. lead the project. X.L and D.K.E prepare the sample. M.K. and U.G. designed the experiment. A.O. and M.K. performed the experiment, discussed the results, and wrote the manuscript. All authors contributed to the preparation of the paper.

## Competing interests

The authors declare no competing interests.
