## [Peer Review File · Communications Physics]

Energy dissipation on magic angle twisted bilayer grapheneReviewers' comments:

Reviewer #1 (Remarks to the Author):

Twisted few layer graphene and other two dimensional materials are hot research candidates for many exceptional properties. The work by Ollier et al measures the nanomechanical energy dissipation of tBLG by a pendulum atomic force microscopy at 5 K. It was found that a strong dissipation signal at $3/4$ band filling appears, which was identified to the Aharonov-Bohm oscillation arising from wavefunction interference between domains of different doping and a signature of orbital ferromagnetism. I have the following suggestions and comments regarding to the work:

1, The authors stated the advantages of p-AFM for the current investigations, however, the authors did not involve any information that connecting the measured signal with the investigated properties. A short description of such connections is necessary to make a better understanding of the method. For example, how the SIS state dissipate (convert) energy to the AFM tip?

2, Fig. 2c shows the twist angle distribution, I am wondering how the twisted bilayer graphene was prepared? Why the twist angle is dependent on locations?

3, How p-type and n-type doped bilayer graphene was formed simultaneously in a sample? How are the domains formed? Is it due to the different contact quality with SiO₂ at different regions?

4, How different domains are separated, i.e., how the white dashed lines in Figs. 2 and 3 are determined.

5, In the current study, the tip was kept 150 nm on top of the samples. How sensitive the measured signal with respect to such a distance? why 150 nm was chosen?

6, In fig. 1, the label for C is missing. The caption for B is also missing. The insert in C has no axis description, number ticks, and units.

7, Two closely related works on t-BLG should be involved in the manuscript: Mater. Today Phys., 2023, 35, 101093; Tuning the lattice thermal conductivity in van-der-waals structures through rotational (dis)ordering. arXiv:2304.06978, doi:10.48550/arXiv.2304.06978.

8, the abstract emphasized to much on the advantage of the p-AFM measurements. I suggest the authors can reduce this part while add more conclusive information on the measured results of t-BLG.

Reviewer #2 (Remarks to the Author):

The manuscript presents and discuss the data of a pendulum atomic force microscopy experiment on a twisted bilayer graphene close to the so-called magic angle.

This technique allows the authors to extract a space-resolved map of the twist-angle, which is rather impressive. The results show a significant twist-angle homogeneity, which is evidently a very important observation. Surprisingly, the dissipation shows oscillations in an applied magnetic field B , especially pronounced at filling $3/4$. The authors attribute the oscillations at large fields in terms of an Aharonov-Bohm interference between currents localised at the boundaries between the hundreds nanometer size domains observed in microscopy, drawing an analogy to the oscillations reported in high-mobility GaAs 2D electron gas. That, if I understand correctly, would imply that at high B the domains behave as p- or n-doped semiconducting ones. On the contrary, the oscillations at small B values are interpreted as due to the magnetism observed at $3/4$ filling, which seems to be mostly of orbital origin.

I find all those results very interesting and the manuscript worth being published in Communications Physics. I have few minor comments to the authors:

(1) The experiment is performed at 5K. If I remember correctly, at this temperature the bilayer becomes metallic at filling $1/2$, while it should be still insulating at $3/4$. Is that the reason why the damping in Fig. 1C is strongly peaked at $3/4$? Maybe some discussion about the value of the temperature at which the experiment is performed and the existing estimates of the gap in the insulating states at various doping might be helpful.

(2) Even though the interpretation of the oscillations in a magnetic field look plausible, I cannot fully understand the behaviour for different fillings. The clearest example is filling $3/4$, where low- and high-B regimes are evidently distinct, fig. 5B. The absence of low-B oscillations at $1/2$ are presumably due to the value of the temperature, see the previous comment. What puzzles me is Fig. 5C for filling above $3/4$. How should one interpret the presence of oscillations for all values of B? Some discussion might be desirable.

(3) In the discussion section the symbol A is used for two different quantity. It is better to use two different symbols.

(4) The panels of the figures are labelled by capital letters but are referred to in the caption by lowercase letter. Maybe using the same in the figure and caption might be better.

In conclusion, if the authors consider the above comments in revised version, I recommend its publication.

Dr. Alexina Ollier
University of Basel
Department of Physics
Klingelbergstrasse 82
4056 Basel
Switzerland

alexina.ollier@unibas.ch

Dear reviewers,

Thank you for your valuable comments. We revised the manuscript accordingly. Below you will find the list of changes into the manuscript.

List of changes:

In the main manuscript:

- The abstract is modified,
- The method section is modified,
- Figure 1 is modified and panels of the figures are labeled by small letters,
- Two new references are included,

In supplementary - Supplementary Section S5 is modified and a new Supplementary Section S8 is included.

Reviewers' comments:

Reviewer #1 (Remarks to the Author):

Twisted few layer graphene and other two dimensional materials are hot research candidates for many exceptional properties. The work by Ollier et al measures the nanomechanical energy dissipation of tBLG by a pendulum atomic force microscopy at 5 K. It was found that a strong dissipation signal at $3/4$ band filling appears, which was identified to the Aharonov-Bohm oscillation arising from wavefunction interference between domains of different doping and a signature of orbital ferromagnetism. I have the following suggestions and comments regarding to the work:

Thank you referee for the fruitful and valuable comments. We will try to address the following points of confusion :

1, The authors stated the advantages of p-AFM for the current investigations,

however, the authors did not involve any information that connecting the measured signal with the investigated properties. A short description of such connections is necessary to make a better understanding of the method. For example, how the SIS state dissipate (convert) energy to the AFM tip?

The description of the pendulum AFM dissipation mechanisms are given in Supplementary section S1.

The dissipation signal Γ is recorded as following: the excitation signal A_{exc} needed to oscillate the tip to a constant amplitude is acquired and converted according to the formula:

$$\Gamma = \Gamma_0 (A_{exc}(d, B, T) / A_0 - f_{exc}(d, B, T) / f_0)$$

where Γ_0 is the dissipation (damping) of the free cantilever. Γ_0 solely depends on the quality of the cantilever structure and external pressure (see : *U. Gysin, S. Rast, P. Ruff, E. Meyer, D.W. Lee, P. Vettger and Ch. Gerber, Temperature dependence of the force sensitivity of silicon cantilevers, PRB69, 045403 (2004)*). In order to minimise Γ_0 the measurements are performed in ultra high vacuum and the cantilevers are annealed to reduce the amount of defects in their structures and to minimise the influence of static charges localized at the tip apex. $A_{exc}(d)$ is the distance (d) dependent excitation amplitude and $f(d)$ is the oscillation frequency of the cantilever that is also distance dependent. The suffix 0 refer to the free cantilever.

The measurements are conducted at large tip sample distances ($d=150\text{nm}$), thus the dominant dissipation channel is long range electrostatic force mediated Joule dissipation (Supplementary section S7 and discussion in the main text), which reads as follows:

$$\Gamma \approx RA^2\omega^2(\Delta\phi)^2(\partial C/\partial z)^2$$

, where R in the sample resistance in the current path, A is the oscillation amplitude, ω is the angular frequency, $\Delta\phi$ is the tip-sample potential controlled by tip voltage (contact potential difference) and $\partial C/\partial z$ the capacitance change between tip and sample.

Since in the experiment A , $\Delta\phi$ (cantilever is grounded), and ω are constant, the only relevant parameters left are R and $\partial C/\partial z$. The resistive response is due to creation of local currents below the oscillating tip and has a form of 'viscous' drag. In this case a tiny amount of localized charges are still present on the tip apex (even after long term annealing of the tip) and the tip vibrations induce an electrical current in the surface region, even if the tip is grounded. The resistive dissipation response leads to wide and smaller peaks in dissipation spectra as visible for $1/4$ and $1/2$ band filling.

The dissipation peaks due to change of system capacitance $\partial C/\partial z$ are sharp and usually show large dissipation values (for $\frac{1}{3}$ and 1 band filling). Their presence is due to charge injection into the flat energy band, which changes thermodynamic compressibility of tBLG and thus implies change of the capacitance of the system. Change of the thermodynamic compressibility is also visible in the frequency shift spectra as shown in Supplementary section S3.

The aspects about the rise of mechanical dissipation for different SIS are discussed in detail at the beginning of the discussion section. In order to make a better link between the measured dissipation signal and different band filling we add a short information in the introduction:

..... Moreover, SIS are detected above an encapsulated device, therefore the tip literally does couple to the existing subsurface phenomena. The rise of dissipation signal is related to the creation of displacement currents under the oscillating tip as well as the change of quantum capacitance of the sample when the charges are injected into the flat energy bands. Both phenomena affect the dynamics of the oscillating tip and lead to the rise of mechanical damping of the cantilever. It is known

2, Fig. 2c shows the twist angle distribution, I am wondering how the twisted bilayer graphene was prepared? Why the twist angle is dependent on locations?

The sample is prepared as described in: *Lu, X., Stepanov, P., Yang, W. et al. Superconductors, orbital magnets and correlated states in magic-angle bilayer graphene. Nature 574, 653–657 (2019). <https://doi.org/10.1038/s41586-019-1695-0>.* We include the reference to this work into Methods section which is now modified:

The tBLG device was fabricated by step-by-step stacking process described in details elsewhere [21]. The hBN/tBLG/hBN/graphite stacks were exfoliated and assembled using a van der Waals assembly technique. First 10nm thick hBN flakes were exfoliated on SiO₂/Si p-doped substrate of resistivity $\rho < 0.005 \Omega \text{cm}$. Next the separated graphene pieces were rotated manually and mechanical cleaning process was applied to release the local strain. Finally tBLG device was capped with 10 nm hBN. The charge concentration was controlled with a DC voltage applied to the p-doped Si backgate, whereas tBLG was coupled capacitively via $d_{\text{SiO}_2} = 300 \text{ nm}$ of SiO₂ oxide and $d_{\text{hBN}} = 10 \text{ nm}$ of hBN . The geometric capacitance of the backgate was equal to:

$$C_{ox} = \frac{C_{\text{SiO}_2} \cdot C_{\text{hBN}}}{C_{\text{SiO}_2} + C_{\text{hBN}}} = 1.11 \cdot 10^{-4} \text{ Fm}^{-2}$$

, where

$$C_{\frac{SiO}{hBN}} = \frac{\frac{\epsilon_{SiO}}{d_{SiO}}}{\frac{\epsilon_{hBN}}{d_{hBN}}}$$

A tBLG device was contacted with 8 golden wires and two of them were grounded during the measurement.

Figure1 and Figure2 are taken from the cited article (links: <https://www.nature.com/articles/s41586-019-1695-0/figures/5>, <https://www.nature.com/articles/s41586-019-1695-0/figures/6>). Figure1 displays a step-by-step stacking process for the fabrication of twisted bilayer graphene (tBLG) with a graphite bottom gate and Figure2 shows images of the fabricated device before and after mechanical cleaning.

Fig1: a–h, Sequential device fabrication method, describing the tear-and-stack co-lamination process used to create the hBN/tBLG/hBN/graphite stacks.

Fig2: a–d, Optical images of the final stacks before mechanical cleaning (a, c) and after mechanical cleaning (b, d).

Concerning the twist angle dependency:

Several studies have shown that the twist angle can strongly vary within the same sample [Zon,Uri]. The interlayer interaction [Yoo], incorporated strain [Huder], surface dislocations [Butz] or structural relaxation due to interactions between graphene and the hBN substrate [Jung] can induce local twist angle variation.

Ref:

- [Zon] Zondiner, U., Rozen, A., Rodan-Legrain, D. et al. Cascade of phase transitions and Dirac revivals in magic-angle graphene. *Nature* **582**, 203–208 (2020). <https://doi.org/10.1038/s41586-020-2373-y>
- [Uri] Uri, A., Grover, S., Cao, Y. et al. Mapping the twist-angle disorder and Landau levels in magic-angle graphene. *Nature* **581**, 47–52 (2020). <https://doi.org/10.1038/s41586-020-2255-3>
- [Yoo] Yoo, H., Engelke, R., Carr, S. et al. Atomic and electronic reconstruction at the van der Waals interface in twisted bilayer graphene. *Nat. Mater.* **18**, 448–453 (2019). <https://doi.org/10.1038/s41563-019-0346-z>
- [Huder] Huder, L. et al. Electronic spectrum of twisted graphene layers under heterostrain. *Phys. Rev. Lett.* **120**, 156405 (2018), <https://doi.org/10.1103/PhysRevLett.120.156405>
- [Butz] Butz, B., Dolle, C., Niekief, F. et al. Dislocations in bilayer graphene. *Nature* **505**, 533–537 (2014). <https://doi.org/10.1038/nature12780>
- [Jung] Jung, J., DaSilva, A., MacDonald, A. et al. Origin of band gaps in graphene on hexagonal boron nitride. *Nat Commun* **6**, 6308 (2015). <https://doi.org/10.1038/ncomms7308>

We modified the following sentence in the main manuscript :

Since the electronic properties of tBLG are extremely sensitive to the homogeneity of the twist angle the cautious control of the stacking process and subsequent cleaning are crucial [21].

3, How p-type and n-type doped bilayer graphene was formed simultaneously in a sample? How are the domains formed? Is it due to the different contact quality with SiO₂ at different regions?

SiO₂ interface is known to trap charges especially under graphene devices [Wu,Zhang]. We mentioned this effect in the our manuscript (ref. 31). The traps typically are a result of sample preparation and the formation of air blisters which would indeed affect the contact quality. To minimise this effect the mechanical cleaning process is necessary, which was performed for our devices (please see point 2).

Another source of domains of different doping is the presence of static charges localized at the SiO₂ oxide interface. The charge-donating impurities below the graphene induce electron density inhomogeneity and generally introduce charge disorder. Charge disorder can be minimised by sample annealing up to few hundred Celcius degree, as the static charges diffuse into the bulk Si. We avoid annealing, as it would lead to high risk of relaxation of the twist angle to random values. Indeed, in our work we noticed good twist angle homogeneity and a significant charge disorder that lead to considerable variation of CPD values.

This presence of charge puddles is typically suppressed by adding an hBN layer in

between the SiO₂ and graphene. hBN screens the random potential fluctuation without compromising the graphene [Lu]. Presumably, in our experiment due to high force sensitivity of the sensor, the bottom hBN bottom layer is not thick enough (10nm) to completely screen the SiO₂ charging effect.

Ref:

[Rhodes] Rhodes, D., Chae, S.H., Ribeiro-Palau, R. et al. Disorder in van der Waals heterostructures of 2D materials. *Nat. Mater.* **18**, 541–549 (2019). <https://doi.org/10.1038/s41563-019-0366-8>

[Wu] Wu, X., Chuang, Y., Contino, A., Sorée, B., Brems, S., Tokei, Z., Heyns, M., Huyghebaert, C., Asselberghs, I., *Adv. Mater. Interfaces* 2018, 5, 1800454. <https://doi.org/10.1002/admi.201800454>

[Zhang] Zhang, Y., Brar, V., Girit, C. et al. Origin of spatial charge inhomogeneity in graphene. *Nature Phys* **5**, 722–726 (2009). <https://doi.org/10.1038/nphys1365>

[Lu] C.-P. Lu, M. R. Vega, G. Li, A. L. Mayer, K. Watanabe, T. Taniguchi, E. Rossi, E. Y. Andrei, *Proc. Natl. Acad. Sci. USA* **113**(24): 6623 (2016), [10.1073/pnas.1606278113](https://doi.org/10.1073/pnas.1606278113)

4, How different domains are separated, i.e., how the white dashed lines in Figs. 2 and 3 are determined.

The comparison of the constant height dissipation maps acquired simultaneously with the frequency shift images (see Supplementary section S5) enables us to determine the different domains. The bright and dark regions visible in the frequency shift images (Figure 6 (C) and (D) in Supplementary section S5) are due to spatial variation of the attractive electrostatic force which is due to different charge concentration. The bright and dark regions in the frequency shift images typically coincide with the non-concentric lines in the dissipation images (white dashed lines).

We add a small comment into the Supplementary Section S5 :

...(c) and (d) are the corresponding frequency shift Δf (Hz) constant height images taken for $n_s/4$ and $n_s/2$, respectively. The bright and dark regions visible in the frequency shift images are due to spacial variation of the attractive electrostatic force, which is due to different charge concentration.

5, In the current study, the tip was kept 150 nm on top of the samples. How sensitive the measured signal with respect to such a distance? why 150 nm was chosen?

The dissipation versus tip-sample distance (d) and doping concentration (n) is shown in figure below. The bright contrast corresponds to the SIS related dissipation peaks. At distances smaller than $d < 100$ nm the position of dissipation peaks changes versus distance in a non-linear fashion due relatively large displacement field induced by the tip. The dissipation intensity also rises, which saturates the input of cantilever

oscillation controller. Therefore, we have chosen to work at larger distances ($d=150\text{nm}$), at which the tip perturbation or tip-induced displacement field is small. The change of the lever arm (marked by white dashed line) is linear and the shift of dissipation peaks versus distance is rather small. At those tip-sample distances of hundreds of nm we are also sure that the dominant energy dissipation channel is Joule dissipation, which is important in regards of data interpretation.

We prepare a new Figure in the Supplementary a Sup. Mat. S8 with the tip-sample distance map:

Figure caption: The energy dissipation map versus doping n and tip-sample distance d . The dissipation peaks are visible as a bright lines at large tip-sample distance (d). The blue dashed line marks the distance d at which series of SIS was detected in dissipation. The white dashed line marks the change of the lever arm, which starts to deviate from linearity at very close tip-sample distances ($d<100\text{nm}$).

6, In fig. 1, the label for C is missing. The caption for B is also missing. The insert in C has no axis description, number ticks, and units.

We thank referee for pointing this mistake. The figure is corrected.

7, Two closely related works on t-BLG should be involved in the manuscript: Mater. Today Phys., 2023, 35, 101093; Tuning the lattice thermal conductivity in van-der-waals structures through rotational (dis)ordering. arXiv:2304.06978, doi:10.48550/arXiv.2304.06978.

Thank you referee for this comment. We added the mentioned articles into the introduction:

During the last decade SIS in tBLG were extensively studied especially in electrical transport conductivity [3, 9], thermal conductivity [Cheng, Eriksson] and capacitance spectroscopy measurements[9, 17].

[Cheng] Yajuan Cheng, Zheyong Fan, Tao Zhang, Masahiro Nomura, Sebastian Volz, Guimei Zhu, Baowen Li, Shiyun Xiong, Mater. Today Phys. 35, 101093 (2023), <https://doi.org/10.1016/j.mtphys.2023.101093>

[Eriksson] Fredrik Eriksson, Erik Fransson, Christopher Linderlv, Zheyong Fan, and Paul Erhart, Tuning the lattice thermal conductivity in van-der-Waals structures through rotational (dis)ordering, arXiv:2304.06978,

doi:10.48550/arXiv.2304.06978.

8, the abstract emphasized too much on the advantage of the p-AFM measurements. I suggest the authors can reduce this part while add more conclusive information on the measured results of t-BLG.

We changed the abstract. Below is the old abstract and the new version:

Old abstract :

“ The traditional Joule dissipation omnipresent in today’s electronic devices is well understood while the energy loss of the strongly interacting electron systems remains largely unexplored. Twisted bilayer graphene (tBLG) is a host to interaction-driven correlated insulating phases, when the relative rotation is close to the magic angle (1.08°). Here, we report on low temperature (5 K) nanomechanical energy dissipation of tBLG measured by pendulum atomic force microscopy (pAFM). The ultrasensitive cantilever tip acting as an oscillating gate over the quantum device shows dissipation peaks attributed to different fractional fillings of the flat energy bands. While conventional transport methods provide quantitative information on correlated insulating states in tBLG, they lack spatial resolution. pAFM, on the other hand, provides spatial resolution and thus allows to determine the twist angle distribution of tBLG. Strikingly it does it without literally touching the sample surface and the tBLG correlated phases are all accessed through the cantilever dynamics without involving any electrical current detection. Application of magnetic fields provoked strong oscillations of the dissipation signal at $3/4$ band filling, which we identified in analogy to Aharonov-Bohm oscillations arising from wavefunction interference present between domains of different doping and a signature of orbital ferromagnetism. The work demonstrates that nano-mechanical energy dissipation provides a rich source of information on the dissipative nature of the correlated electronic system of tBLG, with implications for coupling a mechanical oscillator to the quantum devices. ”

New abstract:

“ The traditional Joule dissipation omnipresent in today’s electronic devices is well understood while the energy loss of the strongly interacting electron systems remains largely unexplored. Twisted bilayer graphene (tBLG) is a host to interaction-driven correlated insulating phases, when the relative rotation is close to the magic angle (1.08°). Here, we report on low temperature (5 K) nanomechanical energy dissipation of tBLG measured by pendulum atomic force microscopy (pAFM). The ultrasensitive cantilever tip acting as an oscillating gate over the quantum device shows dissipation peaks attributed to different fractional fillings of the flat energy bands. Correlated phases are accessed through the cantilever dynamics without touching the sample surface. Local detection allows to determine the twist angle and spatially resolved dissipation images showed the existence of hundred nanometer

domains of different doping. Application of magnetic fields provoked strong oscillations of the dissipation signal at $3/4$ band filling, which we identified in analogy to Aharonov-Bohm oscillations arising from wavefunction interference present between domains of different doping and a signature of orbital ferromagnetism. The work demonstrates that nano-mechanical energy dissipation provides a rich source of information on the dissipative nature of the correlated electronic system of tBLG, with implications for coupling a mechanical oscillator to the quantum devices. ”

Reviewer #2 (Remarks to the Author):

The manuscript presents and discuss the data of a pendulum atomic force microscopy experiment on a twisted bilayer graphene close to the so-called magic angle.

This technique allows the authors to extract a space-resolved map of the twist-angle, which is rather impressive. The results show a significant twist-angle homogeneity, which is evidently a very important observation. Surprisingly, the dissipation shows oscillations in an applied magnetic field B , especially pronounced at filling $3/4$. The authors attribute the oscillations at large fields in terms of an Aharonov-Bohm interference between currents localised at the boundaries between the hundreds nanometer size domains observed in microscopy, drawing an analogy to the oscillations reported in high-mobility GaAs 2D electron gas. That, if I understand correctly, would imply that at high B the domains behave as p- or n-doped semiconducting ones. On the contrary, the oscillations at small B values are interpreted as due to the magnetism observed at $3/4$ filling, which seems to be mostly of orbital origin.

I find all those results very interesting and the manuscript worth being published in Communications Physics. I have few minor comments to the authors:

(1) The experiment is performed at 5K. If I remember correctly, at this temperature the bilayer becomes metallic at filling $1/2$, while it should be still insulating at $3/4$. Is that the reason why the damping in Fig. 1C is strongly peaked at $3/4$? Maybe some discussion about the value of the temperature at which the experiment is performed and the existing estimates of the gap in the insulating states at various doping might be helpful.

We thank the referee for this valuable comment. Indeed $1/2$ filling phase is less robust against the temperature as compared to the other correlated insulating phases [Cao] with a critical temperature and 3.5K and band gap 0.3meV. Our measurement is performed at $T=4.8K$ and that is the reason why the $1/2$ dissipation peak is much less pronounced as compared to the other more robust SIS. The fact that we still are able to observe the response, even at slightly elevated temperatures is most presumably due to the local character of the method. It has been shown that

locally the critical temperatures might show differences as compared to the bulk [Mishina, Kisiel].

[Cao] Cao, Yuan, Valla Fatemi, Ahmet Demir, Shiang Fang, Spencer L. Tomarken, Jason Y. Luo, and others, 'Correlated Insulator Behaviour at Half-Filling in Magic-Angle Graphene Superlattices', *Nature*, 556.7699 (2018), 80–84 <https://doi.org/10.1038/nature26154>

[Mishina] Mishina, E. D., T. V. Misuryaev, N. E. Sherstyuk, V. V. Lemanov, A. I. Morozov, A. S. Sigov, and others, 'Observation of a Near-Surface Structural Phase Transition in SrTiO₃ by Optical Second Harmonic Generation', *Physical Review Letters*, 85.17 (2000), 3664–67 <<https://doi.org/10.1103/PhysRevLett.85.3664>>

[Kisiel] Kisiel, M., F. Pellegrini, G.E. Santoro, M. Samadashvili, R. Pawlak, A. Benassi, and others, 'Noncontact Atomic Force Microscope Dissipation Reveals a Central Peak of SrTiO₃ Structural Phase Transition', *Physical Review Letters*, 115.4 (2015), 046101 <https://doi.org/10.1103/PhysRevLett.115.046101>

We put the following sentence into the manuscript :

..... The external magnetic field provides Zeeman energy $2\mu_B B \approx 0.2\text{meV}$ needed to at least partially close the correlated insulator energy gap, hence the vanishing of dissipation peaks characteristic for different SIS and overall Γ contrast reduction (see Fig.4(c)). **In general, states that occur near half filling are less robust as they have much smaller energy scale when compared to SIS localized at larger band fillings. They are characterized by the band gap equal to about 0.3meV, which corresponds to critical temperature equal to $T_c=3.5\text{K}$ [9]. Larger temperature of our measurement ($T=5\text{K}$) leads to the reduction of dissipation intensity for $\nu = 1/2$ visible in dissipation spectra in Fig.1c, as well as the reduction of magneto-oscillations in Fig. 5a.**

(2) Even though the interpretation of the oscillations in a magnetic field look plausible, I cannot fully understand the behaviour for different fillings. The clearest example is filling 3/4, where low- and high-B regimes are evidently distinct, fig. 5B. The absence of low-B oscillations at 1/2 are presumably due to the value of the temperature, see the previous comment. What puzzles me is Fig. 5C for filling above 3/4. How should one interpret the presence of oscillations for all values of B? Some discussion might be desirable.

We agree with referee on the point that that lack of magnetooscillations at 1/2 filling is presumably due to elevated temperature of the measurement. That observation corresponds to the reduced dissipation intensity at low filling. However, even at 1/2 filling tiny oscillations localized at low B-fields (characteristic for 3/4 filling) are visible. We think it is due to the spatial resolution of pAFM tip which averages over hundred nm area. We comment on that at the end of Results section.

The most probable reason for continuous magneto-oscillations for filling larger than $3/4$ is most presumably similar. Backgate voltage is so close to full filling that the tip mostly senses p- or n-doped semiconducting domains of $\nu = 1$ underneath. The results also show that orbital magnetism is rather sensitive to the doping concentration, since a relatively small rise of band filling above $3/4$ favours the continuous magneto-oscillations.

We modify the manuscript to address this issue :

Results section:

....Therefore, despite the decent twist angle homogeneity measured across micrometer distance, somewhere on the 'horizon' of the tip shadow potential must be domains of different angle twist or different doping. Similar applies to the magneto-oscillation visible at $3/4 < \nu < 1$ FF, when the tip mostly senses p- or n-doped semiconducting domains resulting from close proximity of the full filling.

Hence, the effective sample area probed by the oscillating pendulum tip at $d = 150\text{nm}$ distance must be in the order of few hundred nanometers.

Discussion section:

The observation of magneto-oscillations present at B-field as low as $B < 100\text{mT}$ and localized at $\nu = 3/4$ band filling supports the scenario of chiral edge currents present at domain boundaries, as observed by SQUID measurements. Disappearance of magneto-oscillations at $3/4 < \nu < 1$ FF indicates that network of topologically protected chiral channels is highly sensitive to doping concentration.

(3) In the discussion section the symbol A is used for two different quantity. It is better to use two different symbols.

We changed the letter A for the lateral area to A_L .

(4) The panels of the figures are labelled by capital letters but are referred to in the caption by lowercase letter. Maybe using the same in the figure and caption might be better.

We changed all the figure letters to small letters.

Sincerely yours,
Alexina Ollier, on behalf of the co-authors

REVIEWERS' COMMENTS:

Reviewer #1 (Remarks to the Author):

The authors have carefully revised the manuscript and all my concerns have been addressed appropriately. I therefore recommend the publication of this work on Communication physics.

Reviewer #2 (Remarks to the Author):

I read authors' replies to mine and the first reviewer report, as well as the revised manuscript. I find that the replies and the revised manuscript are very satisfactory, for which reason I advice publication in Communication Physics.

Dr. Alexina Ollier
University of Basel
Department of Physics
Klingelbergstrasse 82
4056 Basel
Switzerland

alexina.ollier@unibas.ch

Dear reviewers,

Thank you for your valuable comments. We revised the manuscript accordingly. Below you will find the list of changes into the manuscript.

List of changes:

In the main manuscript:

- The abstract is modified,
- The method section is modified,
- Figure 1 is modified and panels of the figures are labeled by small letters,
- Two new references are included,

In supplementary - Supplementary Section S5 is modified and a new Supplementary Section S8 is included.

Reviewers' comments:

Reviewer #1 (Remarks to the Author):

Twisted few layer graphene and other two dimensional materials are hot research candidates for many exceptional properties. The work by Ollier et al measures the nanomechanical energy dissipation of tBLG by a pendulum atomic force microscopy at 5 K. It was found that a strong dissipation signal at $3/4$ band filling appears, which was identified to the Aharonov-Bohm oscillation arising from wavefunction interference between domains of different doping and a signature of orbital ferromagnetism. I have the following suggestions and comments regarding to the work:

Thank you referee for the fruitful and valuable comments. We will try to address the following points of confusion :

1, The authors stated the advantages of p-AFM for the current investigations,

however, the authors did not involve any information that connecting the measured signal with the investigated properties. A short description of such connections is necessary to make a better understanding of the method. For example, how the SIS state dissipate (convert) energy to the AFM tip?

The description of the pendulum AFM dissipation mechanisms are given in Supplementary section S1.

The dissipation signal Γ is recorded as following: the excitation signal A_{exc} needed to oscillate the tip to a constant amplitude is acquired and converted according to the formula:

$$\Gamma = \Gamma_0 (A_{exc}(d, B, T) / A_0 - f_{exc}(d, B, T) / f_0)$$

where Γ_0 is the dissipation (damping) of the free cantilever. Γ_0 solely depends on the quality of the cantilever structure and external pressure (see : *U. Gysin, S. Rast, P. Ruff, E. Meyer, D.W. Lee, P. Vettger and Ch. Gerber, Temperature dependence of the force sensitivity of silicon cantilevers, PRB69, 045403 (2004)*). In order to minimise Γ_0 the measurements are performed in ultra high vacuum and the cantilevers are annealed to reduce the amount of defects in their structures and to minimise the influence of static charges localized at the tip apex. $A_{exc}(d)$ is the distance (d) dependent excitation amplitude and $f(d)$ is the oscillation frequency of the cantilever that is also distance dependent. The suffix 0 refer to the free cantilever.

The measurements are conducted at large tip sample distances ($d=150\text{nm}$), thus the dominant dissipation channel is long range electrostatic force mediated Joule dissipation (Supplementary section S7 and discussion in the main text), which reads as follows:

$$\Gamma \approx RA^2\omega^2(\Delta\phi)^2(\partial C/\partial z)^2$$

, where R in the sample resistance in the current path, A is the oscillation amplitude, ω is the angular frequency, $\Delta\phi$ is the tip-sample potential controlled by tip voltage (contact potential difference) and $\partial C/\partial z$ the capacitance change between tip and sample.

Since in the experiment A , $\Delta\phi$ (cantilever is grounded), and ω are constant, the only relevant parameters left are R and $\partial C/\partial z$. The resistive response is due to creation of local currents below the oscillating tip and has a form of 'viscous' drag. In this case a tiny amount of localized charges are still present on the tip apex (even after long term annealing of the tip) and the tip vibrations induce an electrical current in the surface region, even if the tip is grounded. The resistive dissipation response leads to wide and smaller peaks in dissipation spectra as visible for $1/4$ and $1/2$ band filling.

The dissipation peaks due to change of system capacitance $\partial C/\partial z$ are sharp and usually show large dissipation values (for $\frac{1}{3}$ and 1 band filling). Their presence is due to charge injection into the flat energy band, which changes thermodynamic compressibility of tBLG and thus implies change of the capacitance of the system. Change of the thermodynamic compressibility is also visible in the frequency shift spectra as shown in Supplementary section S3.

The aspects about the rise of mechanical dissipation for different SIS are discussed in detail at the beginning of the discussion section. In order to make a better link between the measured dissipation signal and different band filling we add a short information in the introduction:

..... Moreover, SIS are detected above an encapsulated device, therefore the tip literally does couple to the existing subsurface phenomena. The rise of dissipation signal is related to the creation of displacement currents under the oscillating tip as well as the change of quantum capacitance of the sample when the charges are injected into the flat energy bands. Both phenomena affect the dynamics of the oscillating tip and lead to the rise of mechanical damping of the cantilever. It is known

2, Fig. 2c shows the twist angle distribution, I am wondering how the twisted bilayer graphene was prepared? Why the twist angle is dependent on locations?

The sample is prepared as described in: *Lu, X., Stepanov, P., Yang, W. et al. Superconductors, orbital magnets and correlated states in magic-angle bilayer graphene. Nature 574, 653–657 (2019). <https://doi.org/10.1038/s41586-019-1695-0>.* We include the reference to this work into Methods section which is now modified:

The tBLG device was fabricated by step-by-step stacking process described in details elsewhere [21]. The hBN/tBLG/hBN/graphite stacks were exfoliated and assembled using a van der Waals assembly technique. First 10nm thick hBN flakes were exfoliated on SiO₂/Si p-doped substrate of resistivity $\rho < 0.005 \Omega \text{cm}$. Next the separated graphene pieces were rotated manually and mechanical cleaning process was applied to release the local strain. Finally tBLG device was capped with 10 nm hBN. The charge concentration was controlled with a DC voltage applied to the p-doped Si backgate, whereas tBLG was coupled capacitively via $d_{\text{SiO}_2} = 300 \text{ nm}$ of SiO₂ oxide and $d_{\text{hBN}} = 10 \text{ nm}$ of hBN . The geometric capacitance of the backgate was equal to:

$$C_{ox} = \frac{C_{\text{SiO}_2} \cdot C_{\text{hBN}}}{C_{\text{SiO}_2} + C_{\text{hBN}}} = 1.11 \cdot 10^{-4} \text{ Fm}^{-2}$$

, where

$$C_{\frac{SiO_2}{hBN}} = \frac{\frac{\epsilon_{SiO_2}}{d_{SiO_2}}}{\frac{\epsilon_{hBN}}{d_{hBN}}}$$

A tBLG device was contacted with 8 golden wires and two of them were grounded during the measurement.

Figure1 and Figure2 are taken from the cited article (links: <https://www.nature.com/articles/s41586-019-1695-0/figures/5>, <https://www.nature.com/articles/s41586-019-1695-0/figures/6>). Figure1 displays a step-by-step stacking process for the fabrication of twisted bilayer graphene (tBLG) with a graphite bottom gate and Figure2 shows images of the fabricated device before and after mechanical cleaning.

Fig1: a–h, Sequential device fabrication method, describing the tear-and-stack co-lamination process used to create the hBN/tBLG/hBN/graphite stacks.

Fig2: a–d, Optical images of the final stacks before mechanical cleaning (a, c) and after mechanical cleaning (b, d).

Concerning the twist angle dependency:

Several studies have shown that the twist angle can strongly vary within the same sample [Zon,Uri]. The interlayer interaction [Yoo], incorporated strain [Huder], surface dislocations [Butz] or structural relaxation due to interactions between graphene and the hBN substrate [Jung] can induce local twist angle variation.

Ref:

- [Zon] Zondiner, U., Rozen, A., Rodan-Legrain, D. et al. Cascade of phase transitions and Dirac revivals in magic-angle graphene. *Nature* **582**, 203–208 (2020). <https://doi.org/10.1038/s41586-020-2373-y>
- [Uri] Uri, A., Grover, S., Cao, Y. et al. Mapping the twist-angle disorder and Landau levels in magic-angle graphene. *Nature* **581**, 47–52 (2020). <https://doi.org/10.1038/s41586-020-2255-3>
- [Yoo] Yoo, H., Engelke, R., Carr, S. et al. Atomic and electronic reconstruction at the van der Waals interface in twisted bilayer graphene. *Nat. Mater.* **18**, 448–453 (2019). <https://doi.org/10.1038/s41563-019-0346-z>
- [Huder] Huder, L. et al. Electronic spectrum of twisted graphene layers under heterostrain. *Phys. Rev. Lett.* **120**, 156405 (2018), <https://doi.org/10.1103/PhysRevLett.120.156405>
- [Butz] Butz, B., Dolle, C., Niekief, F. et al. Dislocations in bilayer graphene. *Nature* **505**, 533–537 (2014). <https://doi.org/10.1038/nature12780>
- [Jung] Jung, J., DaSilva, A., MacDonald, A. et al. Origin of band gaps in graphene on hexagonal boron nitride. *Nat Commun* **6**, 6308 (2015). <https://doi.org/10.1038/ncomms7308>

We modified the following sentence in the main manuscript :

Since the electronic properties of tBLG are extremely sensitive to the homogeneity of the twist angle the cautious control of the stacking process and subsequent cleaning are crucial [21].

3, How p-type and n-type doped bilayer graphene was formed simultaneously in a sample? How are the domains formed? Is it due to the different contact quality with SiO₂ at different regions?

SiO₂ interface is known to trap charges especially under graphene devices [Wu,Zhang]. We mentioned this effect in the our manuscript (ref. 31). The traps typically are a result of sample preparation and the formation of air blisters which would indeed affect the contact quality. To minimise this effect the mechanical cleaning process is necessary, which was performed for our devices (please see point 2).

Another source of domains of different doping is the presence of static charges localized at the SiO₂ oxide interface. The charge-donating impurities below the graphene induce electron density inhomogeneity and generally introduce charge disorder. Charge disorder can be minimised by sample annealing up to few hundred Celcius degree, as the static charges diffuse into the bulk Si. We avoid annealing, as it would lead to high risk of relaxation of the twist angle to random values. Indeed, in our work we noticed good twist angle homogeneity and a significant charge disorder that lead to considerable variation of CPD values.

This presence of charge puddles is typically suppressed by adding an hBN layer in

between the SiO₂ and graphene. hBN screens the random potential fluctuation without compromising the graphene [Lu]. Presumably, in our experiment due to high force sensitivity of the sensor, the bottom hBN bottom layer is not thick enough (10nm) to completely screen the SiO₂ charging effect.

Ref:

[Rhodes] Rhodes, D., Chae, S.H., Ribeiro-Palau, R. et al. Disorder in van der Waals heterostructures of 2D materials. *Nat. Mater.* **18**, 541–549 (2019). <https://doi.org/10.1038/s41563-019-0366-8>

[Wu] Wu, X., Chuang, Y., Contino, A., Sorée, B., Brems, S., Tokei, Z., Heyns, M., Huyghebaert, C., Asselberghs, I., *Adv. Mater. Interfaces* 2018, 5, 1800454. <https://doi.org/10.1002/admi.201800454>

[Zhang] Zhang, Y., Brar, V., Girit, C. et al. Origin of spatial charge inhomogeneity in graphene. *Nature Phys* **5**, 722–726 (2009). <https://doi.org/10.1038/nphys1365>

[Lu] C.-P. Lu, M. R. Vega, G. Li, A. L. Mayer, K. Watanabe, T. Taniguchi, E. Rossi, E. Y. Andrei, *Proc. Natl. Acad. Sci. USA* **113**(24): 6623 (2016), [10.1073/pnas.1606278113](https://doi.org/10.1073/pnas.1606278113)

4, How different domains are separated, i.e., how the white dashed lines in Figs. 2 and 3 are determined.

The comparison of the constant height dissipation maps acquired simultaneously with the frequency shift images (see Supplementary section S5) enables us to determine the different domains. The bright and dark regions visible in the frequency shift images (Figure 6 (C) and (D) in Supplementary section S5) are due to spatial variation of the attractive electrostatic force which is due to different charge concentration. The bright and dark regions in the frequency shift images typically coincide with the non-concentric lines in the dissipation images (white dashed lines).

We add a small comment into the Supplementary Section S5 :

...(c) and (d) are the corresponding frequency shift Δf (Hz) constant height images taken for $n_s/4$ and $n_s/2$, respectively. The bright and dark regions visible in the frequency shift images are due to spacial variation of the attractive electrostatic force, which is due to different charge concentration.

5, In the current study, the tip was kept 150 nm on top of the samples. How sensitive the measured signal with respect to such a distance? why 150 nm was chosen?

The dissipation versus tip-sample distance (d) and doping concentration (n) is shown in figure below. The bright contrast corresponds to the SIS related dissipation peaks. At distances smaller than $d < 100$ nm the position of dissipation peaks changes versus distance in a non-linear fashion due relatively large displacement field induced by the tip. The dissipation intensity also rises, which saturates the input of cantilever

oscillation controller. Therefore, we have chosen to work at larger distances ($d=150\text{nm}$), at which the tip perturbation or tip-induced displacement field is small. The change of the lever arm (marked by white dashed line) is linear and the shift of dissipation peaks versus distance is rather small. At those tip-sample distances of hundreds of nm we are also sure that the dominant energy dissipation channel is Joule dissipation, which is important in regards of data interpretation.

We prepare a new Figure in the Supplementary a Sup. Mat. S8 with the tip-sample distance map:

Figure caption: The energy dissipation map versus doping n and tip-sample distance d . The dissipation peaks are visible as a bright lines at large tip-sample distance (d). The blue dashed line marks the distance d at which series of SIS was detected in dissipation. The white dashed line marks the change of the lever arm, which starts to deviate from linearity at very close tip-sample distances ($d<100\text{nm}$).

6, In fig. 1, the label for C is missing. The caption for B is also missing. The insert in C has no axis description, number ticks, and units.

We thank referee for pointing this mistake. The figure is corrected.

7, Two closely related works on t-BLG should be involved in the manuscript: Mater. Today Phys., 2023, 35, 101093; Tuning the lattice thermal conductivity in van-der-waals structures through rotational (dis)ordering. arXiv:2304.06978, doi:10.48550/arXiv.2304.06978.

Thank you referee for this comment. We added the mentioned articles into the introduction:

During the last decade SIS in tBLG were extensively studied especially in electrical transport conductivity [3, 9], thermal conductivity [Cheng, Eriksson] and capacitance spectroscopy measurements[9, 17].

[Cheng] Yajuan Cheng, Zheyong Fan, Tao Zhang, Masahiro Nomura, Sebastian Volz, Guimei Zhu, Baowen Li, Shiyun Xiong, Mater. Today Phys. 35, 101093 (2023), <https://doi.org/10.1016/j.mtphys.2023.101093>

[Eriksson] Fredrik Eriksson, Erik Fransson, Christopher Linderlv, Zheyong Fan, and Paul Erhart, Tuning the lattice thermal conductivity in van-der-Waals structures through rotational (dis)ordering, arXiv:2304.06978,

doi:10.48550/arXiv.2304.06978.

8, the abstract emphasized too much on the advantage of the p-AFM measurements. I suggest the authors can reduce this part while add more conclusive information on the measured results of t-BLG.

We changed the abstract. Below is the old abstract and the new version:

Old abstract :

“ The traditional Joule dissipation omnipresent in today’s electronic devices is well understood while the energy loss of the strongly interacting electron systems remains largely unexplored. Twisted bilayer graphene (tBLG) is a host to interaction-driven correlated insulating phases, when the relative rotation is close to the magic angle (1.08°). Here, we report on low temperature (5 K) nanomechanical energy dissipation of tBLG measured by pendulum atomic force microscopy (pAFM). The ultrasensitive cantilever tip acting as an oscillating gate over the quantum device shows dissipation peaks attributed to different fractional fillings of the flat energy bands. While conventional transport methods provide quantitative information on correlated insulating states in tBLG, they lack spatial resolution. pAFM, on the other hand, provides spatial resolution and thus allows to determine the twist angle distribution of tBLG. Strikingly it does it without literally touching the sample surface and the tBLG correlated phases are all accessed through the cantilever dynamics without involving any electrical current detection. Application of magnetic fields provoked strong oscillations of the dissipation signal at $3/4$ band filling, which we identified in analogy to Aharonov-Bohm oscillations arising from wavefunction interference present between domains of different doping and a signature of orbital ferromagnetism. The work demonstrates that nano-mechanical energy dissipation provides a rich source of information on the dissipative nature of the correlated electronic system of tBLG, with implications for coupling a mechanical oscillator to the quantum devices. ”

New abstract:

“ The traditional Joule dissipation omnipresent in today’s electronic devices is well understood while the energy loss of the strongly interacting electron systems remains largely unexplored. Twisted bilayer graphene (tBLG) is a host to interaction-driven correlated insulating phases, when the relative rotation is close to the magic angle (1.08°). Here, we report on low temperature (5 K) nanomechanical energy dissipation of tBLG measured by pendulum atomic force microscopy (pAFM). The ultrasensitive cantilever tip acting as an oscillating gate over the quantum device shows dissipation peaks attributed to different fractional fillings of the flat energy bands. Correlated phases are accessed through the cantilever dynamics without touching the sample surface. Local detection allows to determine the twist angle and spatially resolved dissipation images showed the existence of hundred nanometer

domains of different doping. Application of magnetic fields provoked strong oscillations of the dissipation signal at $3/4$ band filling, which we identified in analogy to Aharonov-Bohm oscillations arising from wavefunction interference present between domains of different doping and a signature of orbital ferromagnetism. The work demonstrates that nano-mechanical energy dissipation provides a rich source of information on the dissipative nature of the correlated electronic system of tBLG, with implications for coupling a mechanical oscillator to the quantum devices. ”

Reviewer #2 (Remarks to the Author):

The manuscript presents and discuss the data of a pendulum atomic force microscopy experiment on a twisted bilayer graphene close to the so-called magic angle.

This technique allows the authors to extract a space-resolved map of the twist-angle, which is rather impressive. The results show a significant twist-angle homogeneity, which is evidently a very important observation. Surprisingly, the dissipation shows oscillations in an applied magnetic field B , especially pronounced at filling $3/4$. The authors attribute the oscillations at large fields in terms of an Aharonov-Bohm interference between currents localised at the boundaries between the hundreds nanometer size domains observed in microscopy, drawing an analogy to the oscillations reported in high-mobility GaAs 2D electron gas. That, if I understand correctly, would imply that at high B the domains behave as p- or n-doped semiconducting ones. On the contrary, the oscillations at small B values are interpreted as due to the magnetism observed at $3/4$ filling, which seems to be mostly of orbital origin.

I find all those results very interesting and the manuscript worth being published in Communications Physics. I have few minor comments to the authors:

(1) The experiment is performed at 5K. If I remember correctly, at this temperature the bilayer becomes metallic at filling $1/2$, while it should be still insulating at $3/4$. Is that the reason why the damping in Fig. 1C is strongly peaked at $3/4$? Maybe some discussion about the value of the temperature at which the experiment is performed and the existing estimates of the gap in the insulating states at various doping might be helpful.

We thank the referee for this valuable comment. Indeed $1/2$ filling phase is less robust against the temperature as compared to the other correlated insulating phases [Cao] with a critical temperature and 3.5K and band gap 0.3meV. Our measurement is performed at $T=4.8K$ and that is the reason why the $1/2$ dissipation peak is much less pronounced as compared to the other more robust SIS. The fact that we still are able to observe the response, even at slightly elevated temperatures is most presumably due to the local character of the method. It has been shown that

locally the critical temperatures might show differences as compared to the bulk [Mishina, Kisiel].

[Cao] Cao, Yuan, Valla Fatemi, Ahmet Demir, Shiang Fang, Spencer L. Tomarken, Jason Y. Luo, and others, 'Correlated Insulator Behaviour at Half-Filling in Magic-Angle Graphene Superlattices', *Nature*, 556.7699 (2018), 80–84 <https://doi.org/10.1038/nature26154>

[Mishina] Mishina, E. D., T. V. Misuryaev, N. E. Sherstyuk, V. V. Lemanov, A. I. Morozov, A. S. Sigov, and others, 'Observation of a Near-Surface Structural Phase Transition in SrTiO₃ by Optical Second Harmonic Generation', *Physical Review Letters*, 85.17 (2000), 3664–67 <<https://doi.org/10.1103/PhysRevLett.85.3664>>

[Kisiel] Kisiel, M., F. Pellegrini, G.E. Santoro, M. Samadashvili, R. Pawlak, A. Benassi, and others, 'Noncontact Atomic Force Microscope Dissipation Reveals a Central Peak of SrTiO₃ Structural Phase Transition', *Physical Review Letters*, 115.4 (2015), 046101 <https://doi.org/10.1103/PhysRevLett.115.046101>

We put the following sentence into the manuscript :

..... The external magnetic field provides Zeeman energy $2\mu_B B \approx 0.2\text{meV}$ needed to at least partially close the correlated insulator energy gap, hence the vanishing of dissipation peaks characteristic for different SIS and overall Γ contrast reduction (see Fig.4(c)). **In general, states that occur near half filling are less robust as they have much smaller energy scale when compared to SIS localized at larger band fillings. They are characterized by the band gap equal to about 0.3meV, which corresponds to critical temperature equal to $T_c=3.5\text{K}$ [9]. Larger temperature of our measurement ($T=5\text{K}$) leads to the reduction of dissipation intensity for $\nu = 1/2$ visible in dissipation spectra in Fig.1c, as well as the reduction of magneto-oscillations in Fig. 5a.**

(2) Even though the interpretation of the oscillations in a magnetic field look plausible, I cannot fully understand the behaviour for different fillings. The clearest example is filling 3/4, where low- and high-B regimes are evidently distinct, fig. 5B. The absence of low-B oscillations at 1/2 are presumably due to the value of the temperature, see the previous comment. What puzzles me is Fig. 5C for filling above 3/4. How should one interpret the presence of oscillations for all values of B? Some discussion might be desirable.

We agree with referee on the point that that lack of magnetooscillations at 1/2 filling is presumably due to elevated temperature of the measurement. That observation corresponds to the reduced dissipation intensity at low filling. However, even at 1/2 filling tiny oscillations localized at low B-fields (characteristic for 3/4 filling) are visible. We think it is due to the spatial resolution of pAFM tip which averages over hundred nm area. We comment on that at the end of Results section.

The most probable reason for continuous magneto-oscillations for filling larger than $3/4$ is most presumably similar. Backgate voltage is so close to full filling that the tip mostly senses p- or n-doped semiconducting domains of $\nu = 1$ underneath. The results also show that orbital magnetism is rather sensitive to the doping concentration, since a relatively small rise of band filling above $3/4$ favours the continuous magneto-oscillations.

We modify the manuscript to address this issue :

Results section:

....Therefore, despite the decent twist angle homogeneity measured across micrometer distance, somewhere on the 'horizon' of the tip shadow potential must be domains of different angle twist or different doping. Similar applies to the magneto-oscillation visible at $3/4 < \nu < 1$ FF, when the tip mostly senses p- or n-doped semiconducting domains resulting from close proximity of the full filling.

Hence, the effective sample area probed by the oscillating pendulum tip at $d = 150\text{nm}$ distance must be in the order of few hundred nanometers.

Discussion section:

The observation of magneto-oscillations present at B-field as low as $B < 100\text{mT}$ and localized at $\nu = 3/4$ band filling supports the scenario of chiral edge currents present at domain boundaries, as observed by SQUID measurements. Disappearance of magneto-oscillations at $3/4 < \nu < 1$ FF indicates that network of topologically protected chiral channels is highly sensitive to doping concentration.

(3) In the discussion section the symbol A is used for two different quantity. It is better to use two different symbols.

We changed the letter A for the lateral area to A_L .

(4) The panels of the figures are labelled by capital letters but are referred to in the caption by lowercase letter. Maybe using the same in the figure and caption might be better.

We changed all the figure letters to small letters.

Sincerely yours,
Alexina Ollier, on behalf of the co-authors